# AMLGENTEX: MOBILIZING DATA-DRIVEN RESEARCH TO COMBAT MONEY LAUNDERING

## ABSTRACT

Money laundering enables organized crime by moving illicit funds into the legitimate economy. Although trillions of dollars are laundered each year, detection rates remain low because launderers evade oversight, confirmed cases are rare, and institutions see only fragments of the global transaction network. Since access to real transaction data is tightly restricted, synthetic datasets are essential for developing and evaluating detection methods. However, existing datasets fall short: they often neglect partial observability, temporal dynamics, strategic behavior, uncertain labels, class imbalance, and network-level dependencies. We introduce AMLGENTEX, an open-source suite for generating realistic, configurable transaction data and benchmarking detection methods. AMLGENTEX enables systematic evaluation of anti-money laundering systems under conditions that mirror real-world challenges. By releasing multiple country-specific datasets and practical parameter guidance, we aim to empower researchers and practitioners and provide a common foundation for collaboration and progress in combating money laundering.

## 1 INTRODUCTION

Money laundering is a critical enabler of organized crime, allowing illicit profits to be integrated into the legitimate economy (Sullivan, 2015; Europol, 2023). The scale of the problem is staggering: an estimated $3.1 trillion in illicit funds flowed through the global financial system in 2023 alone (Nasdaq Verafin, 2024), nearly doubling the $1.6 trillion estimated in 2009 by United Nations Office on Drugs and Crime (UNODC) (2011).

Global anti-money laundering (AML) efforts are anchored in the 2012 FATF Recommendations, which provide a comprehensive framework covering customer due diligence (CDD), including know-your-customer (KYC) procedures, risk profiling, transaction monitoring, reporting through Suspicious Activity Reports (SARs), and record-keeping requirements (Financial Action Task Force (FATF), 2012). However, these recommendations do not prescribe specific implementations, leading to significant variation in how financial institutions (FIs) design risk strategies, CDD processes, and monitoring systems. In practice, most monitoring systems are rule-based.

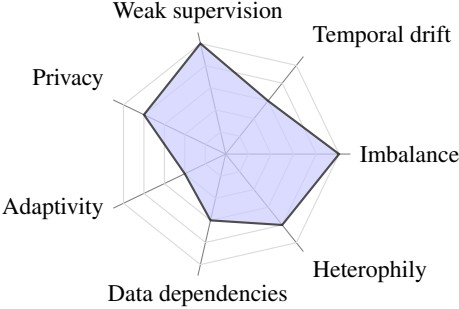

Figure 1: Assessed severity (0–5) of key challenges in transaction monitoring, based on expert opinions from AML practitioners.

Criminals frequently employ sophisticated tactics and rapidly adopt new technologies, often outpacing law enforcement (Europol, 2023). They exploit structural weaknesses, such as the fact that FIs can only observe their internal transaction networks and that data sharing between institutions remains limited (Nasdaq Verafin, 2024; bis, 2023). As a result, AML efforts remain largely ineffective, with success rates estimated as low as 0.2% (Pol, 2020). Compounding these difficulties is the scarcity of open, realistic AML datasets, which significantly hampers research and limits the development and benchmarking of better monitoring systems.

Beyond data limitations, several fundamental factors further amplify the inherent difficulty of AML detection. Through consultations with experienced AML professionals, we identify at least seven key

challenges: (i) **Interdependencies**, since accounts are embedded in a global transaction network that must be considered during monitoring; (ii) **Imbalanced data**, as laundering events are extremely rare compared to legitimate transactions; (iii) **Privacy constraints**, limiting collaboration between institutions (Nasdaq Verafin, 2024); (iv) **Weak supervision**, due to the scarcity of ground-truth labels and the noisy nature of SARs, which often contain unverified or subjective suspicions rather than confirmed illicit activity; (v) **Temporal drift**, as transaction networks naturally evolve over time; (vi) **Adaptive adversaries**, where criminals actively change strategies to evade detection (Europol, 2025); and (vii) **Heterophily**, as launderers deliberately mimic normal behavior, complicating graph-based methods. These challenges, along with their assessed severity, are visualized in Fig. 1.

In this work, we present an open-source data-generation and benchmarking suite called AMLGEN-TEX[1]. Our contributions are threefold. First, we introduce an agent-based simulator combined with an optimization framework for generating transaction networks that capture the challenges outlined above. The optimization can be performed in two modes, depending on the availability of external data. Second, because transaction networks are often difficult to oversee, we provide a visualization tool for seamless inspection and analysis of the generated data. Finally, AMLGENTEX includes a benchmarking suite that incorporates both traditional rule-based methods and recent graph-based detection techniques. To facilitate adoption, we release several country-specific datasets[2] and provide rule-of-thumb guidance with optimization tools for realistic parameter choices. This allows users to generate tailored data and gives practitioners a controlled testbed with ground-truth labels.

## 2 RELATED WORK

Several efforts have been made to synthetically replicate real-world transactional data for AML research. One of the earliest attempts was introduced by Lopez-Rojas & Axelsson (2012), where the authors proposed a purely algorithmic multi-agent simulation focused on mobile payments. Agents probabilistically transitioned between fraudulent and benign behaviors without using any real data. This approach was extended in PAYSIM (Lopez-Rojas et al., 2016), which introduced a reference dataset to empirically estimate agent behavior. However, both works lack spatial structure, as transactions are generated independently across accounts. Moreover, PAYSIM requires an existing real-world dataset to calibrate parameters, which is often unavailable.

To address the lack of spatial components, IBM introduced AMLSIM (Suzumura & Kanezashi, 2021), a multi-agent simulation that does not require a reference dataset. AMLSIM simulates transaction networks based on known AML typologies, categorized into normal and alert patterns. The normal patterns (e.g., direct, mutual, periodic, forward, fan-in, fan-out) and alert patterns (e.g., fan-in, fan-out, cycle, scatter-gather, gather-scatter) reflect typical and suspicious behaviors, respectively. Fig. 8 and Fig. 9 in App. A illustrate these structures. Notably, patterns such as fan-in and fan-out appear in both normal and alert categories, emphasizing how criminals often mimic legitimate activity.

The process of money laundering is often described as consisting of three stages: *placement*, *layering*, and *integration* (Sullivan, 2015; Europol, 2023). AMLSIM primarily focuses on the layering phase. Moreover, it models the transaction network as a closed system with no external inflow or outflow of funds, exhibits inconsistencies between specified and realized patterns, and supports only three of the aforementioned money laundering typologies. Recent efforts, such as (Oztas et al., 2023), have sought to address the latter limitation by expanding the set of normal and alert patterns.

Altman et al. (2023) proposed AMLWORLD, a framework that models relationships between individuals and companies using a circular flow graph. AMLWORLD explicitly simulates the placement, layering, and integration phases. Although the framework itself is not publicly available, the authors have released multiple datasets with varying ratios of alert transactions. However, as mentioned by the authors, the data generation demands extensive parameter tuning and it is unclear how this procedure was performed for the released datasets.

Project Aurora, an initiative by the Bank for International Settlements (bis, 2023), explores the use of privacy-enhancing technologies (PETs) in AML through two months of synthetic transactions generated across six countries and 29 FIs. Although the data generation process is not disclosed, it follows principles similar to those underlying AMLSIM and AMLWORLD.

---

[1] AMLGENTEX is available at `https://anonymous.4open.science/r/AMLGentex-ED23`

[2] Datasets are available at `https://huggingface.co/AMLGentex`

Beyond agent-based simulations, some approaches generate synthetic AML data using statistical models. For instance, SYNTHAML (Jensen et al., 2023) uses Gaussian copulas to produce a synthetic dataset tuned on real transaction data from the Danish bank Spar Nord. The released dataset contains 440,000 clients, 16 million transactions, and 20,000 alert events, and captures entity behaviors such as repeated involvement in money laundering activities.

## 3 PRELIMINARIES

We model the global transaction network at each time $t \in [T]$ as a directed multigraph $\mathcal{G}_t = (\mathcal{V}_t, \mathcal{E}_t, \widetilde{\mathbf{X}}_t, \widetilde{\mathbf{Y}}_t)$. Here, $\mathcal{V}_t$ denotes the set of active accounts at time $t$, where node indices are persistent across time. The multiset of edges is represented as $\mathcal{E}_t = \{(u_i, v_i, a_i)\}_{i=1}^{|\mathcal{E}_t|}$, where each transaction $(u_i, v_i, a_i)$ corresponds to a directed transaction from account $u_i$ to account $v_i$ with associated attributes $a_i \in \mathcal{A}$, such as transaction amount, timestamp, and transaction type. The matrix $\widetilde{\mathbf{X}}_t \in \mathcal{X}_{\text{CDD}}^{|\mathcal{V}_t|}$ contains feature vectors associated with each account, where $\mathcal{X}_{\text{CDD}}$ denotes the input space derived from the CDD process. Moreover, each transaction $(u_i, v_i, a_i)$ is associated with a binary label $\widetilde{y}_i \in \{0, 1\}$ indicating whether the transaction is related to money laundering, and we denote the collection of all labels by $\widetilde{\mathbf{Y}}_t = \{\tilde{y}_i\}_{i=1}^{|\mathcal{E}_t|}$.

Let $w \subseteq [T]$ denote a consecutive window and let $\text{AGG}(w) : \{\mathcal{G}_t\}_{t \in w} \to \mathcal{G}$ be an aggregation operator that combines the graphs over the window $w$ into a single graph $\mathcal{G} = (\mathcal{V}, \mathcal{E}, \mathbf{X}, \mathbf{Y})$ amenable to node classification. The aggregation proceeds as follows: the node set is given by the union $\mathcal{V} = \bigcup_{t \in w} \mathcal{V}_t$, and the edge set by the union $\mathcal{E} = \bigcup_{t \in w} \mathcal{E}_t$. For each node $v \in \mathcal{V}$, we construct a feature vector $\mathbf{x}_v$ by concatenating its static CDD features with aggregated transaction-based features over $w$. The static features are taken from the latest time at which $v$ appears, while the transaction-derived features may include metrics such as total in- and out-flux of funds and the number of unique counterparties. Formally, $\mathbf{x}_v \in \mathcal{X}_{\text{CDD}} \times \mathcal{X}_{\text{TXN}}$, where $\mathcal{X}_{\text{TXN}}$ denotes the space of features derived from transactional activity over the aggregation window. For each node $v \in \mathcal{V}$, we also define the node label $y_v \in \{0, 1\}$ as

$$y_v = \mathbf{1}\left(\exists (u, v, a) \in \mathcal{E} \text{ or } (v, u, a) \in \mathcal{E} \text{ such that } \widetilde{y}_{(u,v,a)} = 1 \text{ or } \widetilde{y}_{(v,u,a)} = 1\right),$$

where $\mathbf{1}(\cdot)$ denotes the indicator function. That is, a node is labeled as money laundering if it appears in a laundering pattern within window $w$, even if it also conducts legitimate transactions.

## 4 DATA GENERATION

The data generation process in AMLGENTEX builds on the multi-agent simulation framework AMLSIM, which we extend by: (i) introducing generalized typologies; (ii) modeling all three stages of money laundering through new placement and integration mechanisms (Europol, 2023); (iii) simulating income and spending beyond a closed network; (iv) enabling control over transaction graph degree distributions; (v) supporting various types of label noise; (vi) incorporating external data sources such as demographic data and public transaction statistics; and (vii) applying systematic hyperparameter tuning to improve data fidelity. A detailed comparison with AMLSIM is given in App. I.1. The generation process is outlined below using private customer accounts within Sweden as illustration. Parameters can be adjusted to model other countries or corporate customers, see App. E.

### 4.1 ACCOUNT RELATIONS

AMLGENTEX initiates by generating a **blueprint network**, where both the number of accounts and their in-degree and out-degree distributions are specified by the user. The distributions follow a scale free pattern, where most accounts have few connections and a small number have many, a structure commonly observed in financial transaction networks (Soramäki et al., 2007; Saxena et al., 2021). The in- and out-degree distributions are realized via a discretized Pareto distribution

$$\mathbb{P}[\deg_u = k] = \frac{1}{\left(\frac{k-\text{loc}}{\text{scale}} + 1\right)^\gamma} - \frac{1}{\left(\frac{k-\text{loc}}{\text{scale}} + 2\right)^\gamma}, \quad k \geq \text{loc}, \quad \gamma > 0$$

with user-provided parameters loc, scale, and $\gamma$ controlling the minimum number of connections, spread, and slope of the distribution. Fig. 10 in App. B illustrates degree distributions under different parameter configurations, and App. E.1 provides guidance on parameter selection.

Next, the blueprint network is populated with user defined typologies from both normal and money laundering scenarios (see Fig. 12, App. B), based on user-specified number, size, and type. Notably, normal patterns are only included if they can be fully accommodated by the blueprint network, while money laundering patterns are injected regardless of fit.

Initially, no account is involved in money laundering and accounts are selected uniformly at random. As typologies are assigned, accounts are grouped according to how many money laundering patterns they have previously participated in. When assigning accounts to new money laundering patterns, at each selection step, with probability $p$, an account is selected from a group with prior participation rather than from the unused pool. Among the participating groups, selection proceeds recursively: with probability $p$, the process moves to a group with a higher participation count, continuing in this way until a group is chosen or the process stops with probability $1 - p$. This mechanism is inspired by real money laundering statistics (Jensen et al., 2023, Fig. 1), where some accounts appear in multiple patterns. The number of money laundering patterns a malicious account participates in, $n_{\mathrm{ml}}$, is modeled by a logarithmic distribution

$$\mathbb{P}[n_{\mathrm{ml}} = k] = \frac{-1}{\log(1 - p)} \cdot \frac{p^k}{k}$$

where $p \in (0, 1)$ is a parameter that controls the probability of reusing accounts. The resulting cumulative distribution function is shown in App. B, Fig. 11, for different values of $p$.

Since normal patterns must fit within the blueprint network, some patterns may be discarded depending on the input configuration. Likewise, due to the random assignments, some accounts may not participate in any pattern after the normal and alert patterns have been applied. These accounts are pruned from the transaction network, which ultimately alters the degree distribution of the resulting network. As a result, the targeted degree distribution may differ from the final degree distribution. The process of creating the relational network is illustrated in App. B, Fig. 12.

## 4.2 ACCOUNT INTERACTIONS

With the account relationships established, the next step is to generate the transactions within each pattern. This is governed by a timing scheme, controlled by user provided start and end times. AMLGENTEX supports several timing schemes: fixed interval, random interval, unordered, and simultaneous. In the fixed interval scheme, transactions occur at regular, user defined intervals. In the random interval scheme, transaction times are sampled within a given time window. The unordered scheme allows transactions to occur at arbitrary times, while in the simultaneous scheme, all transactions within the pattern occur at the same time step. Some typologies also enforce a partial order on transactions. In forward and cycle patterns, transactions follow a strict sequential order. In scatter-gather, gather-scatter, and stacked bipartite patterns, transactions are randomly ordered within each layer, but all transactions in a given layer must complete before any in the next layer can begin.

To model the in-flow of money into the network, we introduce a special **source node**, which connects to all other nodes and distributes monthly incomes. To reflect a realistic distribution of funds, we use external data on mean and median salaries by age group, along with the population age distribution in Sweden (Statistics Sweden, 2024). Ages are represented as integers, and each account is assigned an age via inverse sampling from the empirical cumulative distribution function (CDF) shown in Fig.2a. Given an assigned age, we retrieve the corresponding mean salary $\mu_{\mathrm{salary}}$ and median salary $m_{\mathrm{salary}}$ from(Statistics Sweden, 2024), and model the income using a lognormal distribution:

$$p_{\mathrm{salary}}(x) = \frac{1}{x\sigma_{\mathrm{scale}}\sqrt{2\pi}} \exp\left(-\frac{(\log(x) - \mu_{\mathrm{loc}})^2}{2\sigma_{\mathrm{scale}}^2}\right),$$

with parameters defined as:

$$\mu_{\mathrm{loc}} = \log(m_{\mathrm{salary}}), \quad \sigma_{\mathrm{scale}} = \sqrt{2\log(\mu_{\mathrm{salary}}) - \mu_{\mathrm{loc}}}.$$

Fig. 2b shows the resulting distributions for selected ages $20, 40, 60, 80$. The distribution shifts rightward with age, reflecting increasing income with seniority, and falls off around retirement age.

Analogous to the source node, we introduce a **sink node** to account for the out-flow of funds from the transaction network. The sink node acts as the recipient for all transactions that do not occur between

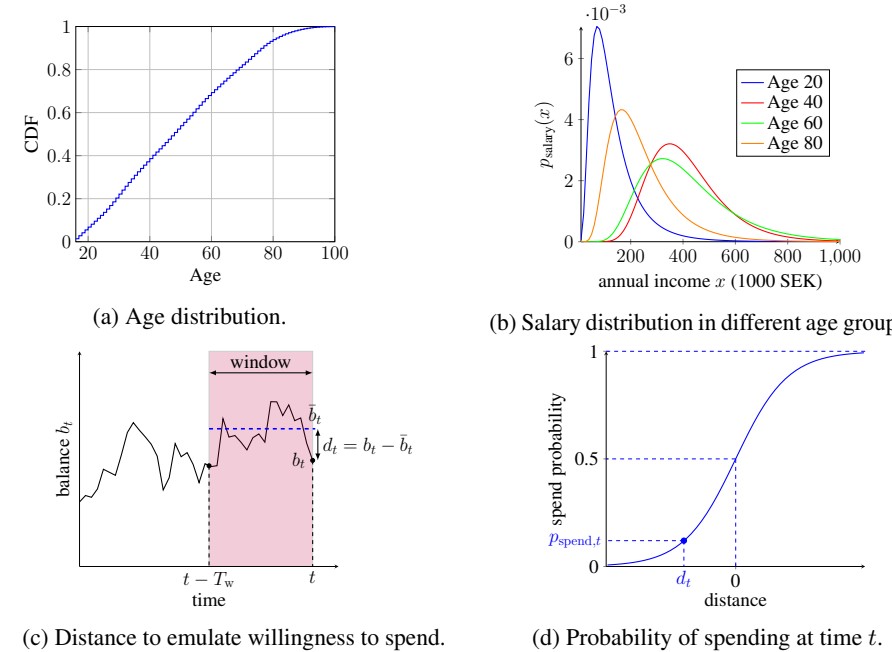

(a) Age distribution.

(b) Salary distribution in different age groups.

(c) Distance to emulate willingness to spend.

(d) Probability of spending at time $t$.

Figure 2: Components used to model in- and out-flow of the transaction network.

two internal accounts, effectively modeling external spending. Spending behavior is simulated using a sliding window of length $T_{\mathrm{w}}$ to compute the average balance $\bar{b}_t$ at time $t$. We then calculate the deviation from this average as $d_t = b_t - \bar{b}_t$, where $b_t$ is the current balance. Intuitively, if $d_t > 0$, the account has a surplus relative to its recent history and is more likely to spend; if $d_t < 0$, spending is less likely. This is modeled probabilistically using a sigmoid function applied to $d_t$, yielding the spending probability at time $t$, denoted $p_{\mathrm{spend},t}$. A transaction to the sink node occurs if a Bernoulli trial with success rate $p_{\mathrm{spend},t}$ returns one. The full process is illustrated in Fig. 2c and Fig. 2d.

All transactions in AMLGENTEX are associated with attributes $a \in \mathcal{A}$, including the transaction amount $x$ and timestamp $t$. The amount is sampled from a truncated Gaussian distribution parameterized by a mean $\mu$, variance $\sigma^2$, a lower limit, and an upper limit. The lower limit is fixed to 1 SEK, representing the smallest possible transaction amount. For each agent, the upper limit is defined as the minimum of the current balance $b_t$ and a maximum allowed amount $x_{\max}$. To distinguish between benign and money laundering transactions, amounts are sampled from separate truncated Gaussian distributions. This allows the difference in transaction behavior to be tuned: for example, the signal strength can be increased by widening the gap between $\mu_{\mathrm{normal}}$ and $\mu_{\mathrm{alert}}$, see Fig. 3.

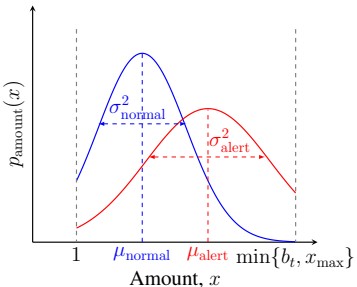

Figure 3: Transaction amount PDFs for a given account at time $t$: normal (blue) vs. laundering (red).

AMLGENTEX models the money laundering process via **placement**, **layering**, and **integration**. Illicit funds can be introduced in two ways: through transfer or cash. In the transfer method, the initiating account receives extra funds along with its salary. These funds are visible to the bank and are used in laundering transactions (see Fig. 4a). In the cash method, the account receives untracked funds before the laundering pattern begins. These funds are stored in a separate cash balance that is not visible to the bank. The cash may originate from criminal activity or from a cooperating end account. Once laundering begins, the cash is used while the bank balance remains unchanged (see Fig. 4b). To reflect real world behavior, the laundering account keeps only a fraction of the received funds (Finanspolisen, 2018). During integration, the account may choose to spend either from cash or from its bank balance. This is modeled as a Bernoulli trial with user defined probability, typically set

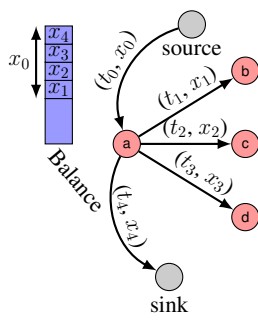 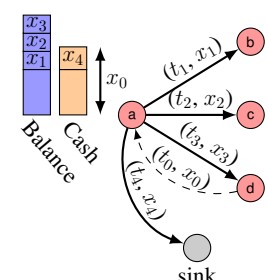

(a) **Placement**: At $t_0$, the source transfers $x_0$ to account a. **Layering**: Account a sends $x_1$, $x_2$, and $x_3$ to b, c, and d, keeping the remainder. **Integration**: At $t_4$, a spends $x_4$ by sending it to the sink.

(b) **Placement**: At $t_0$, account d sends cash $x_0$ to a, invisible to the bank. **Layering**: a sends $x_1$, $x_2$, $x_3$ to b, c, and d, retaining the rest. **Integration**: At $t_4$, a sends $x_4$ to the sink.

Figure 4: Placement, layering, and integration in the different money-laundering procedures.

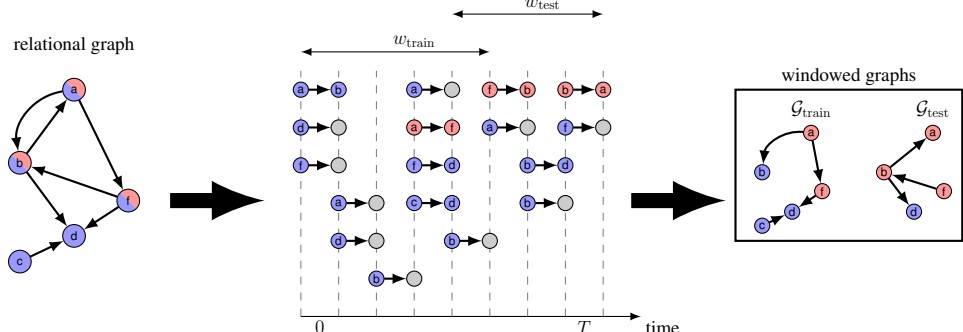

Figure 5: From the relational graph (left), transactions are generated over discrete time-steps. Transactions within the training and testing periods (center) are used to compute account features. The resulting transaction networks are shown on the right with sink transactions omitted.

high due to the decline of cash usage (Sveriges Riksbank, 2024). This behavior may create a weak signal for the FI, as the account's balance remains idle while cash is being spent.

## 4.3 FEATURE ENGINEERING: PROCESSING THE TRANSACTION LOG

After generating the transaction data, a feature processing step is applied to prepare the data for machine learning. Let $t = 0$ and $t = T$ denote the start and end of the transaction log. To support both traditional monitoring approaches (e.g., decision trees) and graph-based methods, the transaction log is split into two user-specified time-windowed graphs: a training graph and a test graph. The training graph is constructed as $\mathcal{G}_{\text{train}} \leftarrow \text{AGG}(w_{\text{train}})$, where AGG aggregates transactions over the window $w_{\text{train}}$. A validation set is obtained by holding out a subset of nodes from $\mathcal{G}_{\text{train}}$. Similarly, the test graph is constructed as $\mathcal{G}_{\text{test}} \leftarrow \text{AGG}(w_{\text{test}})$, where $w_{\text{test}}$ may overlap with $w_{\text{train}}$. This setup is illustrated in Fig. 5. Note that the train and test graphs can also be used with traditional methods by discarding the edges and treating nodes as independent instances.

The node features generated by the aggregation function are listed in Table 1 of App. C, and can be easily extended or modified. To avoid overfitting to the specifics of the data generation process, the feature design is inspired by (bis, 2023, Sec. 8.2) and (Finanspolisen, 2018). To enable fine-grained temporal analysis, the time period is divided into $m$ sub-windows, and each feature is computed separately within each sub-window. As a result, even if an account appears in both $\mathcal{G}_{\text{train}}$ and $\mathcal{G}_{\text{test}}$, its feature representation will likely differ. In addition, as shown in App. D, AMLGENTEX supports injecting label noise to reflect the uncertainty and biases inherent in the labeling process used by FIs.

AMLGENTEX also enables the creation of multiple local transaction networks, each representing a distinct FI. These networks can be configured heterogeneously, with different number of users and typologies, reflecting variation across institutions. This supports the evaluation of local, centralized,

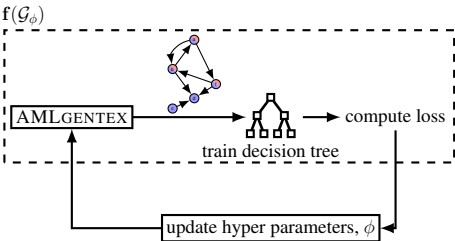
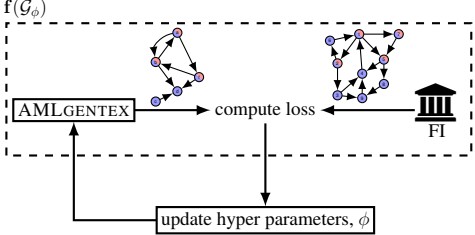

(a) Hyper-parameter tuning towards a target FPR.  (b) Hyper-parameter tuning from real transaction data.

Figure 6: Optimization framework in the knowledge-free setting (a) and data-informed setting (b).

and federated learning scenarios. Importantly, feature vectors are computed using only the information available to each FI. Thus, in local and federated settings, features are derived from the local view of the network, whereas centralized training can leverage global information.

## 5 HYPERPARAMETER TUNING

AMLGENTEX requires the specification of several hyperparameters to generate synthetic transaction data. Setting these parameters is inherently challenging, as many are not publicly disclosed; revealing them could aid money launderers in mimicking benign behavior and evading detection. To address this, we offer two systematic approaches for parameter selection: (i) a knowledge-free setting, where no prior information on money laundering behavior is assumed, targeted towards researchers and collaborative benchmarking without access to sensitive data, and (ii) a data-informed setting aimed towards FIs that can calibrate simulations to internal data. For the knowledge-free setting, practical guidelines on parameter selection are given in App. E.

For both settings, our automatic parameter selection relies on multi-objective Bayesian optimization. We define a vector-valued objective function $\mathbf{f}(\mathcal{G}_\phi) \in \mathbb{R}^k$, where $\mathcal{G}_\phi$ is the generated transaction graph parametrized by the hyperparameters $\phi \in \Phi$ with $\Phi$ denoting the space of available hyperparameters. Since the objectives may be conflicting, the goal is to approximate the Pareto-optimal set

$$\Phi^\star = \{\phi \in \Phi \mid \nexists \phi' \in \Phi, \phi' \neq \phi : \mathbf{f}(\mathcal{G}_{\phi'}) \prec \mathbf{f}(\mathcal{G}_\phi)\},$$

where $\mathbf{f}(\mathcal{G}_{\phi'}) \prec \mathbf{f}(\mathcal{G}_\phi)$ indicates that $\mathbf{f}(\mathcal{G}_{\phi'})$ *Pareto-dominates* $\mathbf{f}(\mathcal{G}_\phi)$, i.e., $\mathbf{f}(\mathcal{G}_{\phi'})$ is no worse in all objectives and strictly better in at least one. Optimization is performed using BOHB (Falkner et al., 2018), which combines Bayesian and bandit strategies, and is implemented via OPTUNA (Akiba et al., 2019), which provides native support for multi-objective optimization.

In the *knowledge-free setting*, parameter selection must rely on publicly available information. For benign transactions in AMLGENTEX, we use aggregate statistics from Swish (Swish, 2023), a mobile payment system, such as the average number of transactions per account and the average transaction amount. Defining parameters for money laundering behavior is more challenging. To address this, we apply the optimization framework described above. We draw inspiration from publicly reported false positive rates (FPR) in rule-based AML systems, which are often as high as 90–98% (PwC US, 2010; McKinsey & Company, 2020). Let $\mathcal{D}_\phi$ denote the individual node features in $\mathcal{G}_\phi$ and let $\mathrm{FPR}(\mathcal{D}_\phi)$ denote the FPR of a decision tree trained on $\mathcal{D}_\phi$, with hyperparameters optimized using OPTUNA. Our first objective minimizes the deviation from an FPR target $\mathrm{FPR}_{\mathrm{target}}$ as

$$f_1(\phi) = |\mathrm{FPR}(\mathcal{D}_\phi) - \mathrm{FPR}_{\mathrm{target}}|.$$

To discourage reliance on single features, we define a second objective that promotes uniform Gini-based feature importance. This creates more challenging benchmarks by removing dominant predictive signals, forcing models to exploit subtler patterns. It represents a deliberate worst-case design choice, and AMLGENTEX allows users to modify, extend, or replace the objectives to align with their own goals. Let $\psi_i(D_\phi)$ be the importance of feature $i \in [N]$, and let $\bar{\psi}(D_\phi)$ denote the mean importance. The second objective is defined as

$$f_2(\phi) = \sum_{i=1}^{N} |\psi_i(\mathcal{D}_\phi) - \bar{\psi}(\mathcal{D}_\phi)|.$$

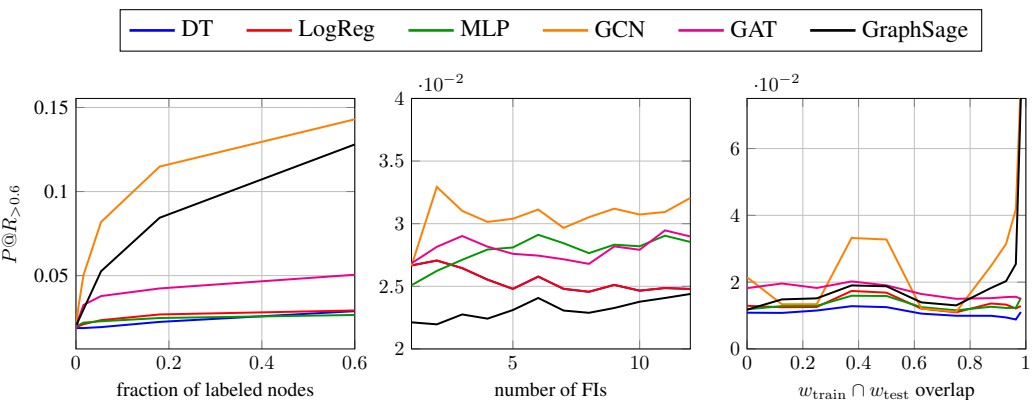

Figure 7: AML monitoring accounting for weak supervision (left), collaborative learning (center), and changing behavior (right).

Finally, we choose $\phi \in \Phi^\star$ closest to the FPR target. The process is illustrated in Fig. 6a.

In the *data-informed setting*, where real transaction data is available, a subset of the real transaction network, $\mathcal{G}_{\text{target}}$, can be used to calibrate the synthetic data by aligning key characteristics. These may include the degree distribution, average in- and out-degrees, and average transaction amounts. Given the sensitivity of such data, FIs can define custom objective functions and tolerances for acceptable similarity. This process is illustrated in Fig. 6b.

# 6 EXPERIMENTS

In this section, we evaluate the performance of traditional methods, e.g., decision trees (DT), logistic regression and multilayer perceptron (MLP), as well as standard graph neural networks for node classification (GCN (Kipf & Welling, 2017), GAT (Veličković et al., 2018), and GraphSAGE (Hamilton et al., 2017)), to study the impact of the challenges outlined in Fig. 1. The dataset is generated using AMLGENTEX in a knowledge-free setting, consists of 100,000 accounts uniformly distributed across 12 FIs over 112 time steps (16 weeks), and is configured with a target FPR of 98%. Static KYC-style features are not included in this simulation, but can be easily appended as demonstrated in App. H. The data is highly imbalanced, with only 1850 accounts related to money laundering activities (see Fig. 15). The class homophily, measured as the proportion of neighbors sharing the same label, equals 0.98 and 0.52 for normal accounts and money launderers, respectively. Hence, most normal accounts connect to other normal accounts, while money laundering accounts are more mixed, suggesting attempts to blend in with normal activity. Dataset parameters, statistics, and generation time are provided in App. F, and details on machine-learning hyperparameters are provided in App. G. Given the importance of minimizing false negatives in AML settings, we report the average precision at recall values above 0.6 as the evaluation metric, denoted $P@R_{>0.6}$. Results are averaged over 10 independent seeds.

**Weak Supervision**: In this experiment, we study the impact of label scarcity in AML. We adopt a transductive setting where $w_{\text{train}} = w_{\text{test}}$, using features constructed from $m = 4$ sub-windows of length 28. For each sub-window, we extract the features described in App. C and concatenate them to form the final representation. We assume a centralized setup where data from all financial institutions is available during training. As shown in Fig. 7 (left), performance declines as label availability decreases, with models varying in sensitivity. Notably, the graph-based methods GCN and GraphSAGE achieve the highest overall performance and benefit most from increased label access, while other models show only modest gains. Given the limited availability of labels in practice, it is crucial to develop models that remain effective under weak supervision.

**Privacy**: In this experiment, we consider the same setting as above but assume access to all node labels during training. Motivated by recent initiatives (bis, 2023; SWIFT, 2025), we investigate the use of federated learning across the 12 FIs, employing Federated Averaging (McMahan et al., 2017), with all FIs using identical hyperparameters as described in App.G. Fig. 7 (center) shows the performance as more FIs join the federation. All models, except logistic regression, show a positive

trend, i.e., it helps slightly to federate with more FIs. However, compared to the centralized setting, where accessing just 60% of the labels yields an average precision close to 0.15 the federated results suggest substantial room for improvement. One promising direction is to explicitly leverage inter-FI connections through subgraph federated learning (Zhang et al., 2021; Aliakbari et al., 2025).

**Changed behavior**: In this section, we consider a scenario where money launderers employ three specific patterns during the first half of the simulated timeline, after which they switch to new typologies. App. F, Fig. 21, illustrates the number of transactions associated with each pattern that emerges in the data. We set $w_{\text{train}} = \{1, \ldots, 56\}$, while the test window $w_{\text{test}}$ is of the same length but varies in position. This setup places us in an inductive setting with a distribution shift between training and test graphs. For simplicity, we assume centralized access to node labels.

Fig. 7 (right) shows the test performance across different degrees of overlap between the training and test windows. As expected, larger overlap improves performance, since the test graph more closely resembles the training graph. Interestingly, the performance shows a nonmonotonic trend: it first declines as the test graph diverges from the training graph, but begins to improve again around 50% overlap. This effect is explained by the nature of the new patterns, specifically *scatter gather* and *gather scatter*, which contain *fan in* and *fan out* substructures at the start of their sequences (see App. A). Furthermore, as shown in Fig. 21, these new patterns account for significantly more transactions. Consequently, the early stage fan in and fan out components of the new patterns become discriminative, temporarily boosting performance. As the latter parts of the typologies dominate, the node representations diverge from the original patterns, leading to another performance drop.

## 7 CONCLUSION

AMLGENTEX offers a simulation suite for generating standardized AML benchmarks, enabling evaluation against a consistent baseline rather than relying on proprietary or limited public datasets. Its configurable simulation engine supports alignment with real world data and targeted method development. In our experiments, we use the framework to assess model robustness to missing labels and temporal drift, and to quantify gains from collaborative detection. Such analyses, grounded in realistic AML scenarios, can guide financial institutions in prioritizing investments, for example, when advanced graph based methods are appropriate or how missing labels affect detection performance. Moreover, empirical evidence of collaborative benefits can inform regulatory discussions and support public and private pilots by providing a neutral testbed for refining data sharing protocols. A more detailed discussion of these implications is provided in App. I.

**Ethical considerations**: Releasing a simulation suite for money laundering data raises legitimate ethical concerns, particularly whether its potential to advance AML research outweighs the risk of misuse. A key question is whether AMLGENTEX could equip criminals with tools to improve their evasion strategies. Importantly, the simulation framework introduces no novel insights into money laundering tactics: typologies, feature sets, and layering strategies are all drawn from documented criminal behavior and prior literature.

The primary theoretical risk lies in enabling adversaries to experiment with data-driven laundering strategies. In practice, however, we consider this risk minimal. The effectiveness of such experimentation is constrained by: (i) the opacity of institutional detection systems, which precludes replication or benchmarking; (ii) the limited fidelity of simulated environments relative to real-world transactional complexity; and (iii) unobservable factors such as labeling practices, risk thresholds, and customer demographics that critically influence detection outcomes. These factors are largely inaccessible to malicious actors and beyond their control. Furthermore, as detection systems grow more diverse and adaptive, adversarial anticipation becomes increasingly infeasible. We view AMLGENTEX as contributing to this trend.

In addition to adversarial threats, a further concern is that of legitimate actors deploying biased or discriminatory monitoring systems as a result of poorly chosen simulation parameters. As with any AI system, responsible use requires explicit risk mitigation strategies, including fairness analysis and bias auditing. Indeed, as AMLGENTEX provides a fully controlled setting for data-generation, it can support such investigations. For example, FIs can deliberately simulate underrepresented or vulnerable populations and evaluate whether models treat them fairly. Conducting such analysis can be difficult with real-world data, where labels are incomplete and potentially subject to the skews of existing monitoring processes.

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

# A  TYPOLOGIES

The normal typologies available are shown in Fig. 8 and the available money laundering are shown in Fig. 9. The fan-in, fan-out, cycle, scatter-gather, gather-scatter, and stacked bipartite patterns can be of arbitrary size set by the user. Moreover, each typology is related to a timing schema chosen by the user.

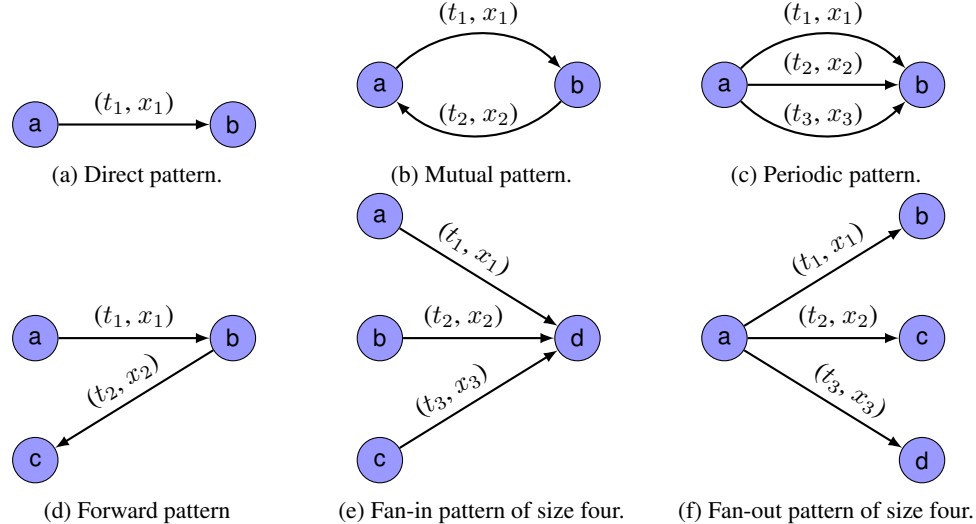

(a) Direct pattern.  (b) Mutual pattern.  (c) Periodic pattern.

(d) Forward pattern  (e) Fan-in pattern of size four.  (f) Fan-out pattern of size four.

Figure 8: Normal patterns. Each transaction is associated with a time stamp $t$ and an amount $x$.

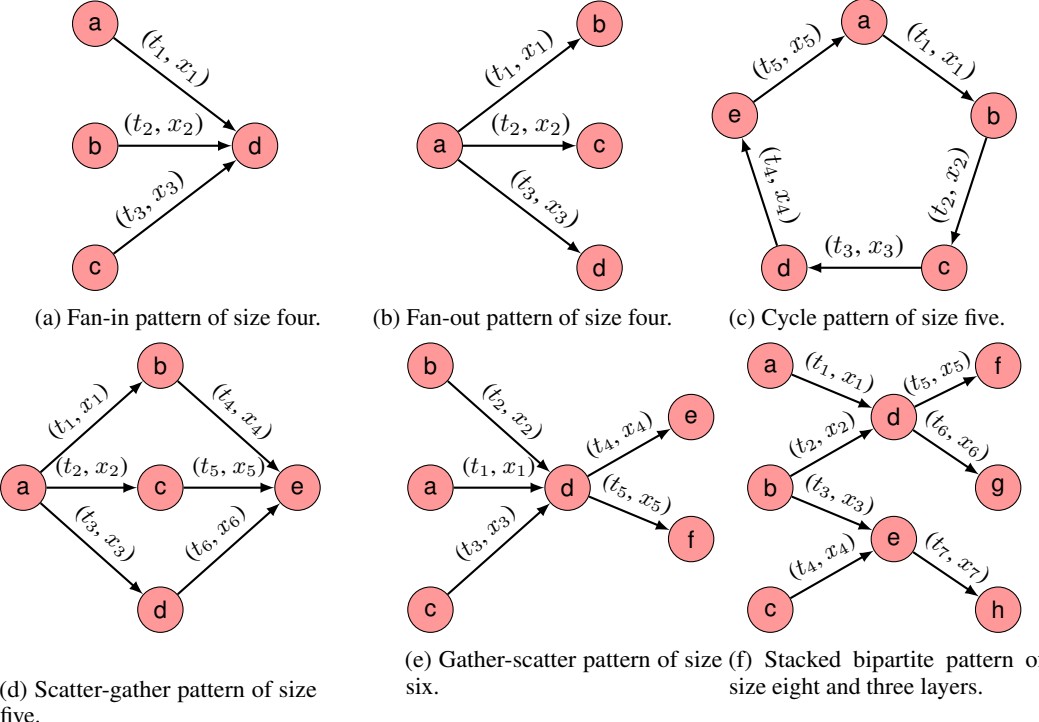

(a) Fan-in pattern of size four.  (b) Fan-out pattern of size four.  (c) Cycle pattern of size five.

(d) Scatter-gather pattern of size five.  (e) Gather-scatter pattern of size six.  (f) Stacked bipartite pattern of size eight and three layers.

Figure 9: Laundering patterns. Each transaction is associated with time stamp $t$ and an amount $x$.

## B  ACCOUNT RELATIONS

The degree distribution of the blueprint network is controlled via a discretized Pareto distribution, parametrized by a triplet $(\mathrm{loc}, \mathrm{scale}, \gamma)$. In Fig. 10, the impact of the parameters are shown. Similarly, the probability of an account engaging in multiple money laundering patterns is given by a logarithmic distribution parameterized on $p$ whose impact is shown in Fig. 11.

The complete process of defining a degree distribution, mapping it to a blueprint network, defining normal patterns and money laundering patterns, and fitting them into the blueprint network is illustrated in Fig. 12.

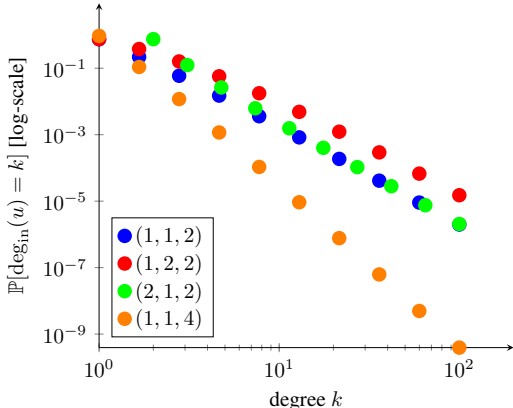

Figure 10: In-degree distribution for different values of $(\mathrm{loc}, \mathrm{scale}, \gamma)$.

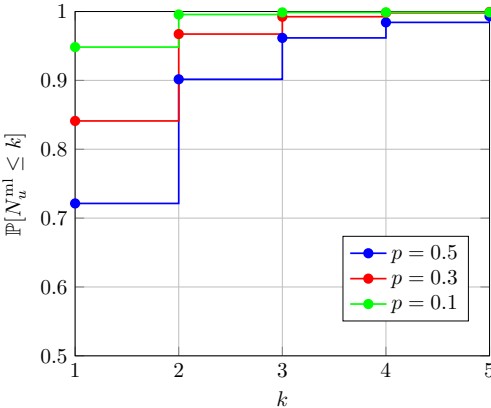

Figure 11: Logarithmic distribution modeling the number of money laundering events illicit accounts engage in.

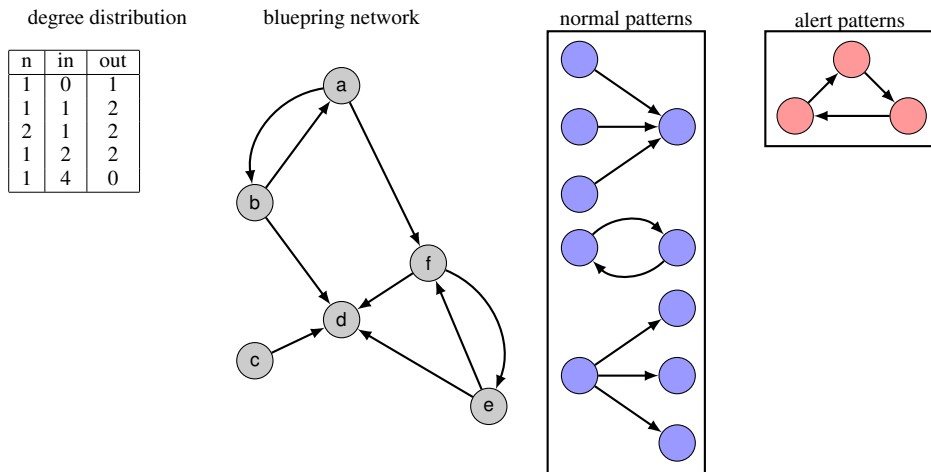

(a) **Preparation of the Transaction Network:** A blueprint network is generated based on the degree distribution. User-defined normal patterns and alert patterns are then created for injection into the blueprint network. In the example above, the normal patterns consist of a fan-in pattern, a mutual pattern, and a fan-out pattern, while the alert pattern is represented by a cycle.

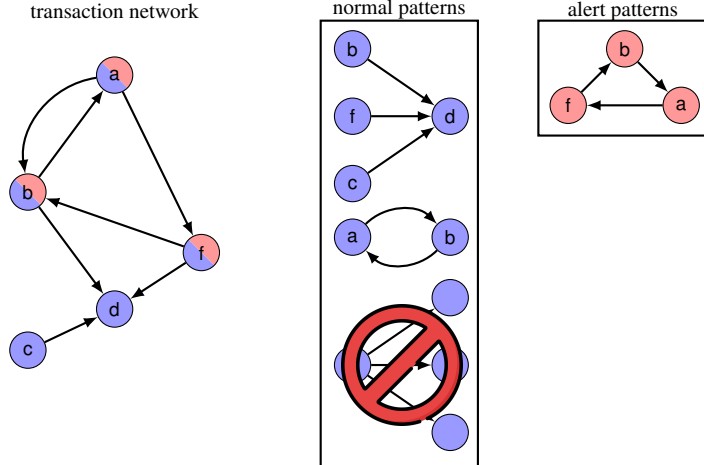

(b) **Pattern injection into the transaction network:** the simulator attempts to locate positions within the blueprint network that can accommodate normal patterns. If no suitable location is found for a given normal pattern, it is discarded and not included in the network. For example, in the case above, the fan-in pattern is mapped to nodes a, b, c, d, while the mutual pattern is mapped to nodes a, b. However, the fan-out pattern is discarded because no account has three outgoing transactions. In contrast, alert patterns are always inserted by randomly selecting nodes according to the distribution in Fig. 11. Specifically, for each node, it is first determined whether a node already involved in illicit transactions should be selected, or if a new node should be sampled uniformly at random from the entire graph. In the example above, nodes a, b, f are chosen for the alert pattern. Since there is no existing link between nodes f and b, a new link will be added to the graph. Additionally, as node e is not involved in any pattern, it will be pruned from the network. In the transaction network above, nodes that belong only to normal or alert patterns are represented as ⬤ and ⬤ , respectively, while nodes that belong to both normal and alert patterns are depicted as ⬤.

Figure 12: Preparation of the transaction network by generating a blueprint based on the degree distribution, followed by the injection of normal patterns (fan-in, mutual, and fan-out) and alert patterns (cycle) into the network. The simulator attempts to fit the normal patterns into suitable nodes, discarding those that cannot be placed, while alert patterns are randomly inserted, creating links between previously unconnected nodes as needed.

## C  FEATURE ENGINEERING

To create the training and test graph, the transaction network is windowed over $w_{\text{train}}$ and $w_{\text{test}}$. Within each window, multiple subwindows are used to collect the transactions within the subwindow and computing the features in Table 1. Notably, these features are inspired by (bis, 2023; Swedish Police Authority, 2020). Once the features for each subwindow has been obtained, they are concatenated to create the feature vector of the corresponding train/test graph.

Table 1: Features created for an account based on its transaction history over a time period.

| Metric | Description |
|---|---|
| sum_spending | Total amount spent by an account. |
| mean_spending | Average amount spent by an account. |
| median_spending | Median amount spent by an account. |
| std_spending | Standard deviation of amounts spent by an account. |
| max_spending | Maximum amount spent in a single transaction. |
| min_spending | Minimum amount spent in a single transaction. |
| count_spending | Total number of outgoing transactions. |
| sum | Total transaction amount for an account. |
| mean | Average transaction amount for an account. |
| median | Median transaction amount for an account. |
| std | Standard deviation of transaction amounts. |
| max | Maximum transaction amount for an account. |
| min | Minimum transaction amount for an account. |
| count_in | Number of incoming transactions. |
| count_out | Number of outgoing transactions. |
| count_unique_in | Number of unique counterparties sending transactions. |
| count_unique_out | Number of unique counterparties receiving transactions. |
| count_days_in_bank | Number of days an account has been active. |
| count_phone_changes | Number of phone number changes for an account. |

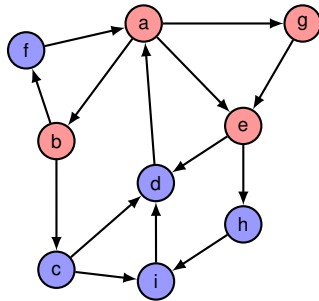
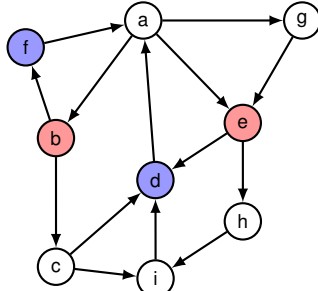

(a) Ground truth data consisting of a single illicit fan-out pattern including accounts $\{a, b, e, g\}$.

(b) Accounts $\{a, c, g, h, i\}$ do not have a label.

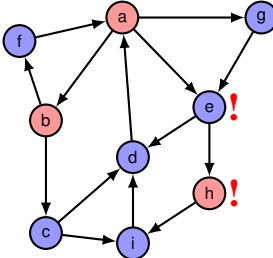
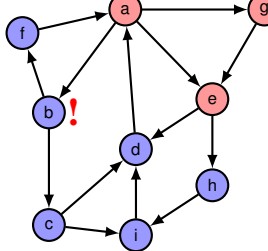
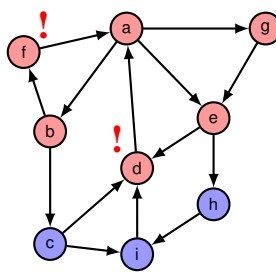

(c) Class noise: benign account $\{h\}$ and suspicious accounts $\{g, e\}$ are mislabeled.

(d) Typology noise: accounts belonging to a given pattern are mislabeled. Here, account $\{b\}$ is falsely flagged as benign.

(e) Neighbour noise: nodes in the vicinity of laundering accounts are mislabeled. Here, account $\{d, f\}$ are flagged as suspicious because of their interactions with $\{a, b\}$ and $\{a, f\}$, respectively.

Figure 13: Toy example illustrating the different types of label noise. Suspicious accounts are denoted as ⬤ and benign nodes as ⬤, respectively. Moreover, mislabeled nodes are denoted by ❗ and nodes without a label are denoted by ◯.

# D   DATA IMPURITIES

Following the creation of $\mathcal{G}_{\text{train}}$ and $\mathcal{G}_{\text{test}}$ from Section 4.3, we introduce a method to create impurities in the data to better reflect reality. In particular, we adopt the view of weakly supervised learning (Zhou, 2018), considering both incomplete labels and inaccurate labels. Unlike fully supervised learning, where each training instance is associated with a precise output label, weakly supervised learning tackles scenarios where acquiring high-quality labels is either expensive or impractical.

Let each account $i$ be represented by a tuple $(\mathbf{x}_i, y_i) \in \mathcal{X} \times \{0, 1\}$, where $\mathbf{x}_i$ is the feature vector of node $i$, constructed as outlined in Section 4.3, $y_i$ is its binary label, and $\mathcal{X} = \mathcal{X}_{\text{XCDD}} \times \mathcal{X}_{\text{XTXN}}$ denotes the feature space. Define $\mathcal{D}_{\text{train}}$ as all the node information of $\mathcal{G}_{\text{train}}$, with index sets $\mathcal{I}_{\text{labeled}}$, $\mathcal{I}_{\text{unlabeled}}$, and $\mathcal{I}_{\text{inaccurate}}$ representing data points with ground truth labels, missing labels, and inaccurate labels, respectively. The training data is then given by

$$\mathcal{D}_{\text{train}} = \bigcup_{i \in \mathcal{I}_{\text{labeled}}} \{\mathbf{x}_i, y_i\} \cup \bigcup_{i \in \mathcal{I}_{\text{unlabeled}}} \{\mathbf{x}_i\} \cup \bigcup_{i \in \mathcal{I}_{\text{inaccurate}}} \{\mathbf{x}_i, \hat{y}_i\}, \qquad (1)$$

where $\hat{y}_i$ denotes an erroneous label for node $i$. Naturally, the supervised setting is recovered for $\mathcal{I}_{\text{unlabeled}} = \mathcal{I}_{\text{inaccurate}} = \emptyset$.

Transaction networks at FIs inherently contain a large number of unlabeled transactions and only a few labeled instances. This scarcity is primarily due to the high cost and complexity of labeling. Each labeled transaction must pass rigorous scrutiny from the FI's compliance teams, requiring in-depth investigations into transaction histories and cross-referencing multiple sources. Moreover, labels generated internally, such as those transactions reported to financial authorities, do not always indicate confirmed cases of money laundering. Even if a reported case proceeds to court and results in

a conviction, which would confirm it as money laundering, feedback on these outcomes rarely reaches the FI promptly, if at all. This lack of timely feedback means that by the time FIs receive confirmation, the patterns involved may already be outdated, as criminals continuously adapt their methods to evade detection. Additionally, privacy regulations restrict FIs from sharing flagged events with one another, limiting opportunities to aggregate and learn from each institution's data. Consequently, labeling remains scarce, costly, and fragmented across the financial industry. In AMLGENTEX, the user may provide a fraction of labeled nodes whereafter the labels in the training data will be pruned accordingly and not used during the training, see Fig. 13b.

Current rule-based systems are known for a high false-positive rate. As investigators get fed with many false positives, it is possible that the transaction gets flagged simply out of caution, resulting in false positive labels for the training. Since this process is largely manual, there are also personal biases involved in the labeling. Moreover, due to the scarce feedback from regulatory authorities, the labels are difficult to verify. Also, as money launderers constantly adapt, it is possible that the monitoring system misses illicit transactions or that the investigator falsely closes a case, resulting in a false negative. In AMLGENTEX, this labeling noise is managed by introducing two types of perturbations: *class noise* and *typology noise*. These add controlled inaccuracies to the training graph, $\mathcal{G}_{\mathrm{train}}$, where the user sets a flip probability for both benign and illicit accounts. Accounts are then randomly flipped according to these probabilities, introducing false positives and false negatives into the training data, as illustrated in Fig. 13c and Fig. 13d.

Since monitoring systems often flag accounts based on network analysis, accounts connected to high-risk entities may also be flagged due to guilt by association. To model this in AMLGENTEX, we introduce *neighbor noise*. As shown in Fig. 13e, accounts connected to known money laundering entities are randomly flagged based on a user-defined probability, simulating the impact of network-based flagging errors.

# E HOW TO CHOOSE HYPERPARAMETERS FOR NORMAL ACCOUNTS

This appendix provides statistics used to model normal accounts before applying Bayesian optimization of the alert transactions. In particular, AMLGENTEX requires (i) the location, scale, and $\gamma$ parameters (see Sec. 4.1), (ii) transaction behavior of normal accounts, and (iii) demographic statistics.

## E.1 HYPERPARAMETERS FOR THE BLUEPRINT GRAPH

To choose $(\text{loc}, \text{scale}, \gamma)$, we consider the mean of the discretized Pareto distribution which is given by

$$\mathbb{E}[\deg] = \text{loc} + \text{scale}\left(\zeta(\gamma) - 1\right), \quad \gamma > 1$$

where $\zeta(\cdot)$ is the Riemann zeta function. This relation provides a practical way to tune the parameters. The location loc sets the minimum degree: $\text{loc} = 0$ allows accounts to remain inactive, while $\text{loc} = 1$ ensures that all accounts have at least one transaction. The tail exponent $\gamma$ controls the heaviness of the distribution: values in $\gamma \in (1, 2)$ produce heavy-tailed distributions and are often unrealistic, $\gamma \in [2, 3]$ are typical of scale-free networks where a few hubs coexist with many low-degree nodes, and $\gamma > 3$ leads to a faster decay and more homogeneous degree distributions (Albert & Barabási, 2002, Table II). Finally, scale adjusts both the spread and the mean of the distribution. Given a target average degree $d$, one can backsolve as

$$\text{scale} = \frac{d - \text{loc}}{\zeta(\gamma) - 1}.$$

Hence, in the knowledge-free setting, one may use canonical values such as $\text{loc} = 1$ and $\gamma = 2.5$, while the average degree $d$ can be chosen according to the observed average number of transactions per account (see Table 2). With these parameters, scale follows directly and the degree distribution of the blueprint graph is determined.

## E.2 HYPERPARAMETERS FOR NORMAL ACCOUNT BEHAVIOR

Table 2: Mobile payment vendor statistics: user base, average number of payments per user per month ($d$), average transaction value ($\mu_{\text{normal}}$), and maximum amount allowed ($x_{\text{max}}$).

| vendor | region | users | $d$ | $\mu_{\text{normal}}$ | $x_{\text{max}}$ |
|---|---|---|---|---|---|
| Zelle | US | 151M | 4 | \$284 | \$10000 |
| Venmo | US | 91–97M | $\sim$5 | \$65–75 | \$10000 |
| Bizum | Spain | 28.2M | $\sim$3.2 | €40.4 | €1000 |
| BLIK | Poland | 19.4M | $\sim$13 | PLN 149 | PLN 20000 |
| Swish | Sweden | 8.7M | $\sim$4 | 637 SEK | 150000 SEK |
| TWINT | Switzerland | 6M | $\sim$11 | CHF 47 | CHF 5000 |

In Table 2, we provide statistics drawn from a combination of vendor-published reports and secondary industry analyses. Whenever possible, we rely on official disclosures from the payment providers themselves:

- **Zelle (US):** Data are from Early Warning Services, the operator of Zelle, which publishes quarterly statistics on user activity, transaction volumes, and transaction sizes (Early Warning Services, LLC, 2025). Note that the maximum transaction size depends on the corresponding bank.

- **Venmo (US):** PayPal, the parent company of Venmo, reports total payment volumes and overall user base in quarterly filings. However, it does not disclose average transaction values or per-user activity. These metrics are therefore taken from secondary industry sources, e.g., Business of Apps, which aggregate PayPal disclosures and estimate per-user metrics (Business of Apps, 2025).

- **Swish (Sweden):** The operator of Swish publishes detailed monthly statistics including number of private users, average number of payments per user, and transaction sizes, segmented by private, business, and commerce payments (Swish, 2023).

- **BLIK (Poland):** Official press releases from BLIK provide active user counts, total number of transactions, and average transaction values. From these, we compute the average number of transactions per user per month (BLIK / Polski Standard Płatności, 2025).

- **Bizum (Spain):** Bizum publishes annual press notes with the total number of users, number of operations, and total transaction value. From these official totals we derive both the average number of transactions per user per month and the average transaction size (Bizum, 2025).

- **TWINT (Switzerland):** TWINT press releases disclose user numbers and total transaction counts. From these we compute the average number of transactions per user per month. TWINT does not publish average transaction values; instead, we use the Swiss Payment Monitor (Graf et al., 2023) which reports the average value of mobile payments in Switzerland, a reasonable proxy given TWINT's market dominance (TWINT AG, 2025).

These statistics provide direct input for configuring the normal accounts and the blueprint network. Specifically, the number of users determines the total number of accounts, while the average transaction size $\mu_{\text{normal}}$ and the maximum transaction size $x_{\max}$ define the transaction amount distributions, see Fig. 3. In addition, the average degree $d$ can be used to determine the scale parameter once the loc and $\gamma$ parameters have been specified.

Table 3: Population share in different countries by age group (%).

|  | 16–19 | 20–24 | 25–34 | 35–44 | 45–54 | 55–64 | 65+ |
|---|---|---|---|---|---|---|---|
| United States | 6.7 | 6.6 | 13.8 | 13.4 | 12.2 | 12.6 | 17.8 |

|  | <18 | 20–24 | 25–49 | 50–64 | 65+ |
|---|---|---|---|---|---|
| Poland | 5.0 | 4.8 | 36.0 | 18.6 | 20.5 |
| Spain | 5.4 | 5.4 | 33.5 | 22.1 | 20.4 |
| Switzerland | 5.0 | 5.3 | 34.6 | 20.9 | 19.3 |

Table 4: Median monthly earnings in different countries by age group (local currency, 2024).

|  | 16–19 | 18–24 | 25–34 | 35–44 | 45–54 | 55–64 | 65+ |
|---|---|---|---|---|---|---|---|
| United States (USD) | 2560 | 3128 | 4556 | 5404 | 5520 | 5216 | 4164 |

|  | <18 | 20–24 | 25–49 | 50–64 | 65+ |
|---|---|---|---|---|---|
| Poland (PLN) | 4332 | 4131 | 4810 | 4721 | 4128 |
| Spain (EUR) | 1373 | 1528 | 1631 | 1732 | 1671 |
| Switzerland (CHF) | 3752 | 4158 | 4465 | 4761 | 3371 |

### E.3 HYPERPARAMETERS FROM DEMOGRAPHIC STATISTICS

Finally, to model the in-flow of funds into the transaction network, AMLGENTEX requires salary distributions across accounts. Tables 3 and 4 report demographic statistics in terms of population share by age group and median salaries per age group, respectively. For the United States, median salary data are taken from (U.S. Bureau of Labor Statistics, 2025) and demographic shares from (U.S. Census Bureau, 2025). For European countries, median salaries are obtained from (Eurostat, 2025b), and demographic statistics from (Eurostat, 2025a).

# F  EXPERIMENTAL DATA

In this section, we illustrate the generated dataset based on the method presented in 6a. The degree distribution, see Section B, is generated using $\gamma = 2.0$, $\mathrm{loc} = 1.0$, and $\mathrm{scale} = 1.0$. We attempt to fit 80,000 instances of each normal pattern and allow sizes between three and ten for fan-in and fan-out patterns (the remaining patterns are fixed in size).

## F.1  FINDING THE HYPERPARAMETERS FOR ALERT ACCOUNTS

To decide the parameters going into the dataset generation, we fix the parameters for the normal patterns as shown in the *Normal* column of Table 5 by using publicly available information from the Swish mobile payment system (Swish, 2023). We then optimize the money laundering parameters, i.e., the mean and variance of the transaction size distribution, the probability of spending cash if available, the probability to participate in multiple alert patterns, and the parameters going into the phone and bank changes, respectively, by means of Bayesian optimization using OPTUNA. As an objective, we make use of the average precision for recall larger than $0.6$, and target an FPR of $0.98$ when forced to use a decision tree classifier. This value is chosen based on reported values from the literature and aims to create a dataset that is very challenging for classical methods, see, e.g., (PwC Luxembourg, 2024). The resulting parameters for the money laundering patterns are shown in Table 5 under column *Money laundering*.

Table 5: Parameter values for transactions within different normal and alert typologies. The normal parameters are selected from (Swish, 2023) whereas the money laundering parameters are optimized.

| Parameter | Normal | Money laundering |
|---|---|---|
| mean amount | 637 | 799 |
| standard deviation amount | 300 | 163 |
| max amount | 150000 | 150000 |
| min amount | 1.0 | 1.0 |
| mean outcome | 500 | 328 |
| standard deviation outcome | 100 | 105 |
| mean phone change | 1460 | 1272 |
| standard deviation phone change | 365 | 281 |
| mean bank change | 1460 | 1335 |
| standard deviation bank change | 365 | 368 |
| probability of multiple alert patterns | - | 0.218 |
| probability of spending cash | - | 0.147 |

To create the final transaction graphs, we employ the parameters in Table 5 and run the simulation for 112 time-steps corresponding to 16 weeks. We then let $w_{\mathrm{train}} = w_{\mathrm{test}} = [0, 112]$. That is, the training and test period overlap for all of the data. Hence, after creating the feature representations, the train and test graph contain the same nodes and edges but differ in the labels considered. To create the features, we consider $m = 4$ sub-windows, each of size 28, and compute the values in Table 1 for each window.

## F.2  DATA STATISTICS

In Fig. 14, we show the degree distribution of the dataset for FI 1 along with the estimated parameters in the Pareto distribution.

In Fig 15, the node (account) and edge count for the transaction network at each FI is displayed alongside the centralized setting (all). It can be seen that the the $98.15\%$ of the accounts are benign whereas $99.72\%$ of the transactions are benign. Hence, out of the $100000$ accounts, only $1850$ accounts engage in money laundering.

Focusing on the money laundering patterns, in Fig. 16, it can be seen that patterns occur across multiple FIs with some patterns spanning up to seven FIs. Moreover, it can be seen that the alert patterns vary in size ranging between 5 and 20 accounts, sometimes containing more than 40 transactions within the pattern.

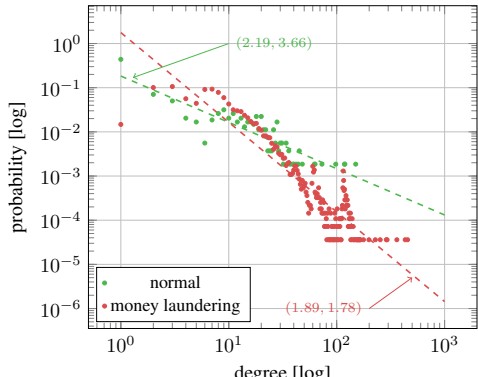

Figure 14: Degree distribution for FI A (left), FI B (middle) and FI C (right). The dashed lines correspond to fitted lines with parameters $(\gamma, \text{scale})$ as shown in the figure.

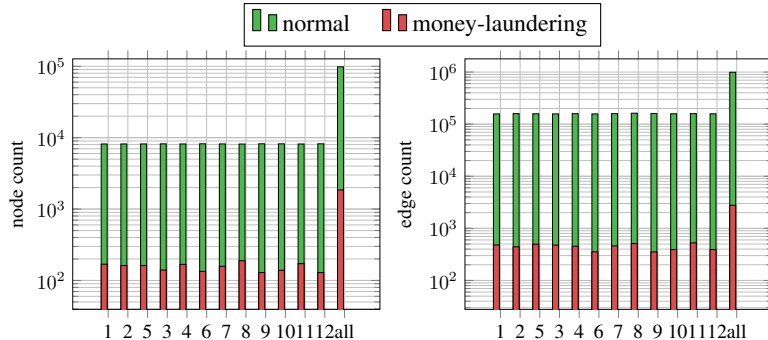

Figure 15: Node (left) and edge (right) counts of positive and negative labels for the generated datasets across FI 1 to FI 12.

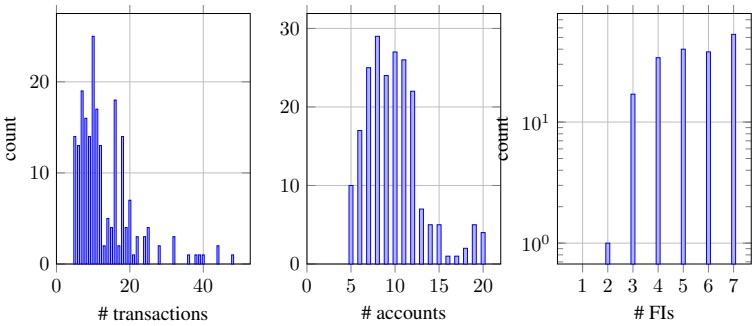

Figure 16: Overview of money laundering typologies in the global transaction graph. Specifically, the plots show: (left) the number of transactions associated with each typology, (middle) the number of accounts involved, and (right) the number of financial institutions spanned by each pattern.

Next, considering FI 1, Fig. 17 illustrates the empirical distributions of transaction amounts and the number of transactions for both normal and money-laundering accounts. From the first row, we observe that the empirical distributions of transaction size and transaction count differ only slightly between the negative and positive classes. However, the distributions for sink transactions (bottom row) appear similar. This suggests that transaction amount and transaction count contain some signal relevant to distinguishing between the two classes but a monitoring algorithm must look beyond.

The pattern counts for FI 1 are shown in Fig. 18 for the different datasets. It can be seen that the transaction network consists of a plethora of different typologies. In Fig. 20, we demonstrate three

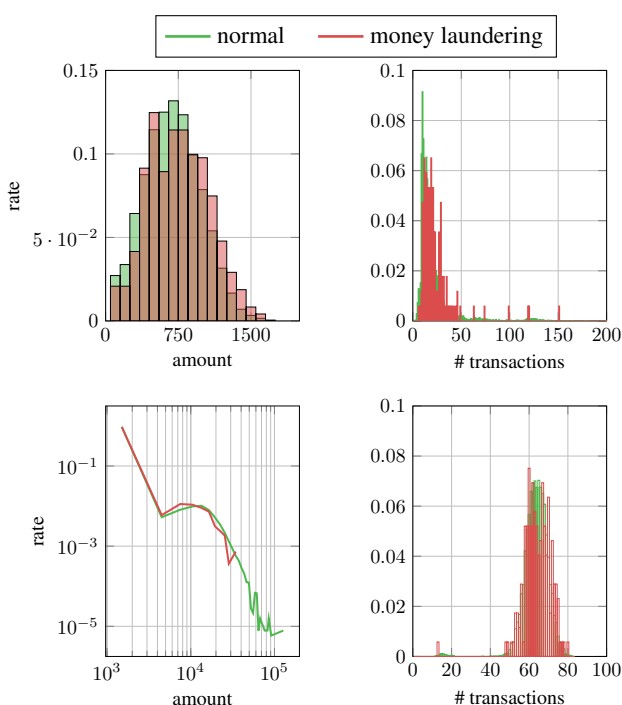

Figure 17: Empirical distributions within a single financial institution: (upper left) distribution of individual transaction amounts; (upper right) number of account-to-account transactions; (lower left) cumulative distribution of the total amount sent to the sink per account; and (lower right) number of transactions to the sink per account.

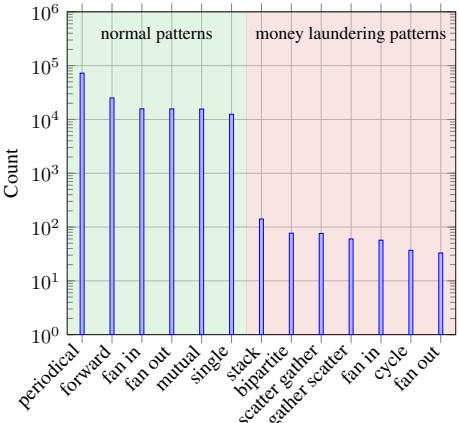

Figure 18: Different patterns in the local transaction network of FI A. Normal patterns are shown with a green back ground and alert patterns are shown with a red background. The count for all three datasets are included in the figure.

randomly chosen money laundering patterns and the transactions that accounts involved engage in. As can be seen, the patterns are of varying complexity and the money launderers involve also in normal transactions.

In Fig. 19, we illustrate some randomly sampled account balances from FI 1 for all time steps in the dataset. It can be seen that each account exhibits a highly individual behavior.

Finally, in Fig. 21, we illustrate the money laundering transactions related to our third experiment in Section 6 pertaining to changed behaviour. It can be seen that the money launderers engage in different typologies within the first and second half of the dataset.

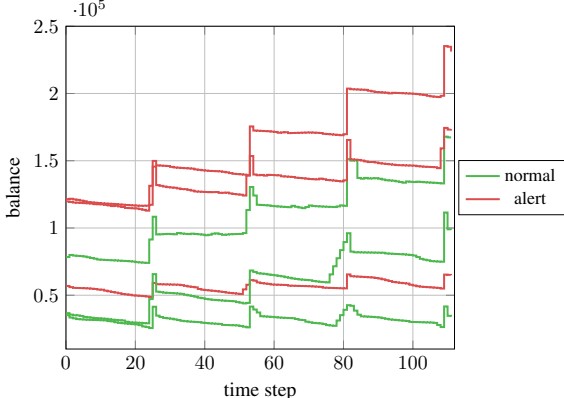

Figure 19: Account balances of three non-laundering accounts and three laundering accounts.

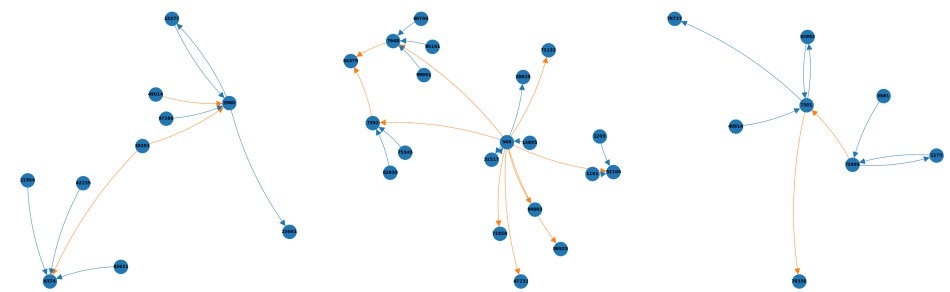

Figure 20: Example patterns from the dataset. Orange links refer to money laundering transactions and blue links refer to normal transactions.

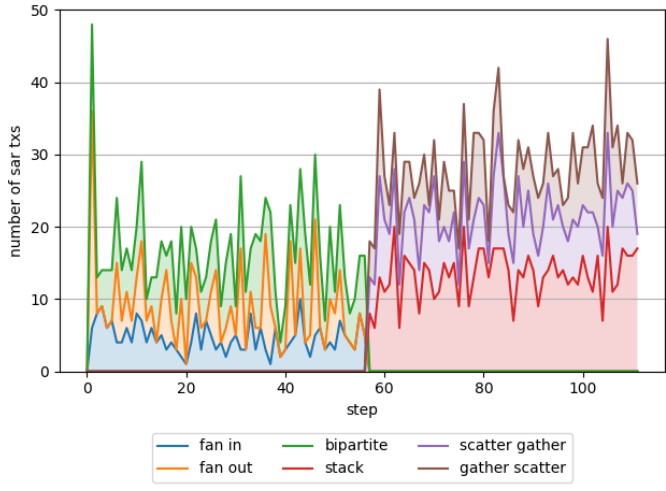

Figure 21: Prevalence of different money laundering transactions and what patterns they belong to. Within time steps 1 to 56, only fan-in, fan-out, and bipartite typologies are utilized. After step 56, the typologies are changed to stack, scatter-gather, and gather-scatter.

## F.3 RUN-TIME

Along with the paper, we release datasets for multiple countries, each in two sizes: 100k and 1M accounts. The 100k-account datasets (112 time steps) were generated in approximately 295 seconds on a standard workstation equipped with an Intel(R) Core(TM) i9-10900X CPU @ 3.70 GHz and 44 GB RAM. Under the same conditions, generating the 1M-account datasets required about 4062 seconds.

These timings reflect a single generation of the transaction log, assuming that hyperparameters have already been determined. If Bayesian optimization is performed as part of dataset generation, the total runtime scales proportionally with the number of optimization iterations.

## F.4 HYPERPARAMETERS FOR OTHER DATASETS

Along with the paper, we release datasets for three mobile payment systems: Swish (Sweden), Venmo (United States), and Bizum (Spain). The Swedish dataset is presented in detail in App. F.2. For the Venmo and Bizum datasets, we set the graph and normal account behavior parameters following the procedure described in App. E. Bayesian optimization is then performed in the data-free setting, as outlined in Sec. 5, and the resulting parameters are reported in Table 6 and Table 7.

Table 6: Venmo dataset parameters. The normal parameters are selected based on the Venmo statistics in Table 2 and the US demography data in Tables 3–4 whereas the money laundering parameters are optimized under the data-free setting. Amounts are in USD.

| Parameter | Normal | Money laundering |
|---|---|---|
| mean amount | 70 | 76 |
| standard deviation amount | 35 | 42 |
| max amount | 10000 | 10000 |
| min amount | 1.0 | 1.0 |
| mean outcome | 60 | 41 |
| standard deviation outcome | 30 | 24 |
| mean phone change | 1460 | 1283 |
| standard deviation phone change | 365 | 332 |
| mean bank change | 1460 | 1272 |
| standard deviation bank change | 365 | 413 |
| probability of multiple alert patterns | - | 0.139 |
| probability of spending cash | - | 0.334 |

Table 7: Bizum dataset parameters. The normal parameters are selected based on the Bizum statistics in Table 2 and the Spanish demography data in Tables 3–4 whereas the money laundering parameters are optimized under the data-free setting. Amounts are in Euro.

| Parameter | Normal | Money laundering |
|---|---|---|
| mean amount | 40.4 | 80 |
| standard deviation amount | 20 | 39 |
| max amount | 1000 | 1000 |
| min amount | 1.0 | 1.0 |
| mean outcome | 30 | 41 |
| standard deviation outcome | 15 | 19 |
| mean phone change | 1460 | 1264 |
| standard deviation phone change | 365 | 330 |
| mean bank change | 1460 | 1277 |
| standard deviation bank change | 365 | 315 |
| probability of multiple alert patterns | - | 0.085 |
| probability of spending cash | - | 0.279 |

# G  MODEL PARAMETERS

For all experiments in Section 6, we use the same hyper parameters. In particular, for the first two experiments, we consider the transductive setting, i.e., the train and test graphs are the same. Assuming access to node-labels, we obtain the hyperparameters of each model by means of Bayesian optimization. The search space and obtained parameters for both the centralized and the federated settings are displayed in Table 8.

During model training, we employ the binary cross-entropy loss function, the ADAM optimizer, early stopping, a bath-size of 512 and train for 300 epochs.

Table 8: Bayesian optimization search space and the obtained values.

| model | parameter | search range | centralized | federated |
|---|---|---|---|---|
| DT | criterion | [gini, entropy, log loss] | log loss | - |
| DT | splitter | [best, random] | random | - |
| DT | max depth | [1, 50] | 7 | - |
| GAT | dropout | [0.2, 0.9] | 0.3449 | 0.2912 |
| GAT | hidden dimension | [100, 300] | 272 | 283 |
| GAT | learning rate | [0.0001, 0.1] | 0.0115 | 0.0617 |
| GAT | layers | [1, 5] | 4 | 1 |
| GCN | dropout | [0.2, 0.9] | 0.2125 | 0.2029 |
| GCN | hidden dimension | [100, 300] | 228 | 103 |
| GCN | learning rate | [0.0001, 0.1] | 0.0499 | 0.0543 |
| GCN | layers | [1, 5] | 2 | 1 |
| GraphSAGE | dropout | [0.2, 0.9] | 0.2075 | 0.3633 |
| GraphSAGE | hidden dimension | [100, 300] | 168 | 270 |
| GraphSAGE | learning rate | [0.0001, 0.1] | 0.0001 | 0.0011 |
| GraphSAGE | layers | [1, 5] | 5 | 1 |
| LogisticRegressor | learning rate | $[10^{-5}, 0.01]$ | 0.0052 | 0.0100 |
| MLP | hidden dimension | [50, 300] | 294 | 259 |
| MLP | learning rate | $[10^{-5}, 0.01]$ | 0.0000 | 0.0017 |
| MLP | hidden layers s | [1, 5] | 1 | 1 |
| MLP | weight decay | [0.0, 1.0] | 0.0044 | 0.0005 |

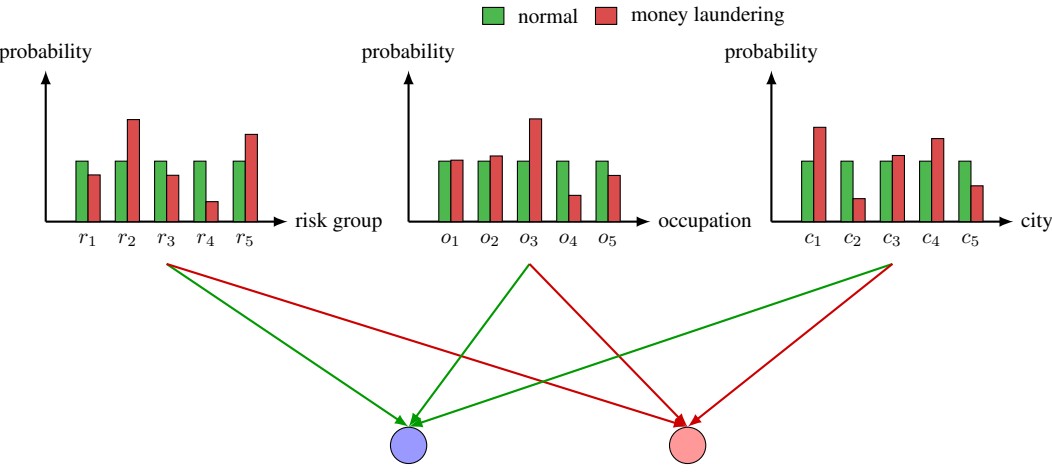

Figure 22: Normal accounts ⬤ and money-laundering accounts ⬤, respectively sample KYC features from different probability distributions defined over the same support.

## H  KNOW-YOUR-CUSTOMER ATTRIBUTE GENERATION

Recommendation 10 in (Financial Action Task Force (FATF), 2012) require FIs to identify and verify customers, understand the purpose and intended nature of business relationships, and apply a risk-based approach to ongoing monitoring. To reflect these principles, AMLGENTEX supports the generation of synthetic KYC-style attributes resembling those used in CDD processes.

AMLGENTEX can produce user-defined categorical features such as age group, region, job sector, employment status, and income bracket. These attributes are excluded from the released datasets due to bias and sensitivity concerns, but remain fully implemented in the open-source codebase. Sampling is distribution-driven: normal accounts draw values independently from uniform distributions over each attribute domain, whereas alert accounts can be skewed toward particular sectors, regions, or demographic groups to simulate risk-driven correlations. An illustration of this process is provided in Fig. 22. This setup enables experimentation with CDD-aligned KYC features, supporting analysis of risk-based patterns and their effect on downstream AML models.

# I DISCUSSION

## I.1 COMPARISON TO ALTERNATIVES

Existing open-source AML simulators are scarce. AMLSIM (Suzumura & Kanezashi, 2021) provided an important early contribution, while AMLGENTEX extends the idea through a complete re-implementation that broadens the scenarios that can be modeled. In contrast, AMLWORLD is not open source, which makes it difficult to assess its design choices, assumptions, or suitability for benchmarking. Table 9 summarizes key differences between AMLSIM and AMLGENTEX.

Table 9: Comparison of AMLSim and AMLGENTEX.

| Dimension | AMLSIM | AMLGENTEX |
|---|---|---|
| Network structure | Multi-institution, closed network | Multi-institution, open network with in- and out-flows |
| Phases modeled | Limited layering focus | Full placement–layering–integration pipeline |
| Demographics / KYC | Random assignment | Configurable attributes (age, region, sector, income) linked to laundering risk |
| Income and spending | Not modeled | Linked to public demographic statistics |
| Transaction patterns | Limited set, several patterns not fully implemented | Extended set of benign and illicit patterns |
| Scheduling | Predefined, simple cycles | More complex temporal schemes |
| Multiple laundering events | Not supported | Accounts may participate in several laundering instances over time |
| Label noise | Not supported | Class, typology, neighbor noise, and missing labels |
| Benchmarking | Basic use cases | Weak supervision, temporal drift, federated learning |
| Feature engineering | Transaction logs only | Modules for both graph-based and non-graph data representations |
| Parameter tuning | Manual configuration | Automated tuning in knowledge-free and data-informed modes; configurable Bayesian optimization with user-defined objectives; rule-of-thumb defaults provided for key parameters |
| Testing | Limited | Extensive unit tests and visualization tools |
| Validation against real data | Not validated | Calibrated and stress-tested against proprietary datasets in partner banks |

In summary, AMLGENTEX generalizes the ideas introduced in AMLSIM into a more comprehensive and extensible platform. Its ability to simulate multiple institutions, more complex typologies and scheduling, repeated laundering events, and richer demographic structures, combined with automated parameter tuning through a configurable Bayesian optimization framework, makes it a practical tool for both research and applied model development. At the same time, rule-of-thumb defaults are provided for key parameters such as network size, degree distributions, and normal account behavior, lowering the entry barrier for new users while retaining the flexibility required for advanced experimentation.

## I.2 LIMITATIONS

AMLGENTEX provides a controlled approximation of reality, and several limitations should be noted. First, the current release focuses on private individual or corporate accounts separately. This reflects a natural scoping of the problem, since FIs typically treat private and corporate accounts as distinct domains with different risk profiles, KYC practices, and product portfolios. Modeling both jointly would therefore be inconsistent with how AML monitoring is organized in practice, and introduce an excessive level of complexity to the problem. At the same time, the framework allows parameters such as network structure, income distributions, transaction volumes, and typology participation to be adjusted to better approximate individual or corporate behavior, and entities can be interpreted as either individual or corporate depending on the configuration.

Second, AMLGENTEX does not include cross-border transactions. This omission is deliberate, as modeling normal account behavior across jurisdictions would require extensive national statistics on income, transaction norms, and typology preferences. For example, cash usage is considered atypical in Sweden (Sveriges Riksbank, 2024) but is common in Germany (Deutsche Bundesbank, 2024), leading to very different laundering signatures. Introducing such heterogeneity would drastically increase the number of parameters, making the framework more complex to configure and less accessible.

Third, the current design treats FIs homogeneously in terms of KYC processes. While it is possible to assign different network structures to different institutions by varying typologies, we do not simulate institution-specific timing patterns or product portfolios. As with cross-border modeling, this choice reflects a trade-off: capturing such heterogeneity would require many additional parameters and would complicate reproducibility, especially in the knowledge-free setting.

A further limitation concerns fidelity in the data-free setting. For this setting, AMLGENTEX is intended as a research platform and the generated datasets should not be interpreted as faithful replicas of production data. Instead, they provide a configurable environment where assumptions are explicit, parameters can be tuned, and models can be stress-tested. To support this use case, in App. E, we provide rule-of-thumb defaults for network size, and degree distributions together with an optimization module that can tune parameters toward a desired monitoring performance. This Bayesian optimization is flexible and allows alternative objective functions to be defined by the user, enabling adaptation to different research questions. In practice, this makes the framework a useful sandbox for academic research even if its fidelity to specific FIs may be limited.

## I.3 OUTLOOK

Beyond technical extensions, the framework also supports transparency and reproducibility in line with regulatory expectations such as the EU AI Act, under which AML may be considered a high-risk application. By making assumptions explicit and experiments auditable, AMLGENTEX provides a controlled environment where detection methods can be systematically compared, and where explainability techniques can be stress-tested in a safe and reproducible way. In this sense, the framework not only advances benchmarking but also enables research on explainability and transparency, which are central requirements for future AML systems. Just as benchmark datasets in computer vision accelerated progress despite abstraction, we believe AMLGENTEX can play a similar role for AML research by offering a shared basis for experimentation and collaboration between academia, regulators, and financial institutions.

Concretely, the controlled environment of AMLGENTEX allows researchers and practitioners to isolate specific factors and probe phenomena that are difficult to observe empirically, even if the simulated data cannot fully capture the complexities of real-world laundering. This includes, for example, studying the impact of missing or noisy labels (as demonstrated in Section 6), assessing model resilience to sudden shifts in transaction behavior, and analyzing sensitivity to demographic composition or rare typologies. Through careful adjustment of demographics, degree distributions, typologies, and transaction patterns, targeted analyses such as coverage of rare laundering strategies or the propensity to over- or under-flag specific groups can be conducted. The resulting insights can inform deployment strategies such as threshold setting and model specialization, while also supporting bias, fairness, and transparency audits that are increasingly required for high-risk AI applications such as AML.

