# OpenReview forum: "AMLGENTEX: Mobilizing Data-Driven Research to Combat Money Laundering"
_ICLR.cc/2026/Conference — ICLR 2026 Conference Withdrawn Submission_

### Official Review · Reviewer_R4zk · 2025-10-27

**Soundness:** 3
**Presentation:** 3
**Contribution:** 2
**Rating:** 2
**Confidence:** 4

**Summary:**

This paper proposes a platform called AMLGENTEX, which integrates multimodal data and large-scale model (LLM) capabilities related to anti-money laundering (AML) in the financial sector. It attempts to apply technologies such as RAG, graph neural networks, and agent frameworks to real-world compliance scenarios. Overall, this paper is a systems integration-oriented, engineering-oriented paper, emphasizing practicality and cross-institutional collaboration.

**Strengths:**

1. Money laundering identification is a high-risk, cross-border, and cross-modal data governance challenge. The paper's inclusion of this as an AI research topic is highly significant.

2.This paper constructs a processing platform, encompassing multimodal data indexing, graph construction, agent interaction, and model-driven development, with a complete overall pipeline.

**Weaknesses:**

1.	This article is more like a systems engineering report or project white paper and does not offer original technical contributions to the AML detection algorithm, model optimization, or learning mechanism. For example: No clear algorithmic formula derivation, No new loss or model structure, RAG partially reuses standard techniques.
2.	Experimental validation is very limited. The experimental section primarily consists of graphs, lacking benchmark-based quantitative results (such as comparisons with traditional AML systems, LLM/RAG improvement technologies, etc.), making the system's superiority less convincing.
3.	The paper's positioning is unclear. If this paper is submitted as an ICLR research track paper, its model-level contribution is unclear. If it is submitted as an industry track paper, additional experimental and engineering performance analysis are needed.

**Questions:**

1. Does the system retain intermediate outputs and task chains for auditability? In high-stakes scenarios like AML, how do you ensure reliability and robustness against LLM-specific risks such as hallucination or prompt injection?
2. Is there any evidence that multi-modal RAG performs better than standard text-only retrieval in the AML context?
3. How is the AML knowledge graph schema defined? Does it rely on a fixed ontology for entities and relations, or is it adaptive across different jurisdictions and regulatory frameworks? Is there support for schema-free or dynamic KG construction?
4. Have you conducted any real-world evaluations comparing AMLGENTEX to existing AML detection systems (e.g., rule-based, GNN-based, or other commercial tools)?

---

### Official Review · Reviewer_o4FY · 2025-10-30

**Soundness:** 3
**Presentation:** 2
**Contribution:** 2
**Rating:** 4
**Confidence:** 4

**Summary:**

This paper introduces AMLGentex, a new agent-based method to generate synthetic data, including a time-varying patterns. It generates dynamic spending profiles for each node in the network, where the authors show how to make these look like country-specific and real-world profiles. Given the network, normal and illicit patterns are injected.

This paper's main contribution is the release of a novel generator for money laundering data that can be tweaked such that the actors' transactions have similar distributions to real-world priors.

**Strengths:**

The authors have provided an extensive method for constructing synthetic transaction networks that can be used to test AML methods.
The payment patterns change over time, allowing to mimic cyclical behaviour during the months.

The flexibility that is built-in using the hyperparameters give the user the ability to alter distributions to their specific needs. This last point is also illustrated in the paper using some country-specific statistics.

The network also includes a source and sink node, allowing for money to enter an leave the system. This gives more flexibility and mimics real-life payment dynamics.

**Weaknesses:**

In general, I have the feeling that the presentation of the method in the main text is a bit too qualitative. To fully understand what the authors mean or how the method is implemented, I had to go to the appendix often. I am of the opinion that the main text should be self-contained enough for the reader to understand the main points of the method.

The text itself is also vague in arguing what the added value is. This is not clearly stated in the introduction, and also the conclusion is a bit vague on this point. Additionally, the related work including Appendix I.1 only compare AMLGentex to AMLSim (Table 9). This ties into a more general remark, since I have the feeling that the related work does not discuss how this work fits into the wider literature. It is my opinion that broader discussions are needed to compare this method to synthetic data in general and say how this helps to forward the AML literature.

Related to that, I believe a more extended reflection on the implications of this work should be discussed in the conclusion.
Deep in the appendix, the authors have put the limitations of this work. I find this an important part of the reflection that should be made in the main body of text.

Some claims are made at the end of the introduction, but I believe these are missing. There is not clear visualisation tool introduced. Additionally, a short discussion on some experiments are included, but to me this doe snot constitute a benchmarking suite, as is claimed in the introduction. I also believe that the experiments are missing some crucial methods. For classification, XGBoost is the state-of-the-art, but it is missing from the experiments. Also, only one evaluation metric is reported, but other well-established metrics should be added, like the area under the precision-recall curve.

**Questions:**

I don't fully understand the need to include Figure 1. Also, how are these opinions gathered? The set-up of this experiment and the analysis of the results seem very opaque to me.

How is the initial balance, b0, set for the nodes in the network? Is this also based on age?

From the text, it is not fully understand how the target FPR is calculated when generating the network. Can the authors elaborate on this?

---

### Official Review · Reviewer_71ox · 2025-10-31

**Soundness:** 3
**Presentation:** 3
**Contribution:** 2
**Rating:** 4
**Confidence:** 4

**Summary:**

The paper presents AMLgentex, an open-source agent-based simulator and benchmarking suite for anti-money laundering (AML). It extends AMLSim by modeling all three laundering stages (placement, layering, integration), supporting open-network flows via source/sink nodes, adding label-noise mechanisms, and providing automated hyperparameter tuning. It offers a knowledge-free mode that targets public statistics, and a data-informed mode that calibrates to institutional characteristics. The experiments examine weak supervision, federated learning, and temporal distribution shift.

**Strengths:**

- Clear parameter-tuning strategy via multi-objective Bayesian optimization with two modes (knowledge-free and data-informed).
- The open-source release intent and detailed hyperparameter configurations promote reproducibility.
- Paper is clearly written and easy to follow.

**Weaknesses:**

- The work extends AMLSim and aims to address some of its limitations; however, several of these have already been resolved in more recent simulators (e.g., AMLWorld by Altman et al.). Comparisons should therefore include these newer data generation models.
- The abstract overstates the shortcomings of existing datasets.
- The uniform feature-importance (Section 5) objective lacks ablations or justification.
- The baselines in the experimental evaluation do not consider more sophisticated AML-specific or directed multigraph architectures.
- Evaluation focuses solely on P@R>0.6; additional metrics such as PR-AUC and F1 are needed given the strong class imbalance.
- Implementation details of the baselines and experimental setup are insufficiently described. No variance or standard deviation is reported.

**Questions:**

- Normal patterns are only retained if the blueprint network can accommodate them, and the nodes uninvolved in any pattern are pruned. What is the reason for requiring all nodes to belong to a pattern? Does this pruning introduce bias or alter network properties?
- How do the authors ensure that random injection of alert patterns does not distort topological properties in a way that makes “alert” nodes trivially distinguishable from normal ones?
- How does having a single source/sink node affect graph connectivity? Specifically, does the presence of a single source/sink node imply that every node in the graph becomes connected through it?
- Could you clarify the main purpose of the time-windowing process? From my understanding, it enables fine-grained temporal features by segmenting transactions into smaller sub-windows, but this seems more related to the solution space (i.e., how models capture temporal dynamics) rather than a dataset-level characteristic.
- Could the authors provide more details on the experimental setup? Specifically, what motivated the choice of a transductive setting rather than an inductive one? Additionally, could you clarify the proportions of nodes used in the train, validation, and test splits?
- Please see weaknesses.

---

### Official Review · Reviewer_8o9c · 2025-11-01

**Soundness:** 2
**Presentation:** 2
**Contribution:** 2
**Rating:** 2
**Confidence:** 4

**Summary:**

This paper presents AMLGENTEX, an open-source framework for generating synthetic transaction datasets for anti-money laundering (AML) research. The system extends previous work (particularly AMLSIM) by using agent-based simulation combined with Bayesian optimization to create transaction networks that capture important challenges in AML detection: partial observability, class imbalance, weak supervision, temporal drift, and network dependencies.

This paper addresses an important practical problem and provides a comprehensive engineering solution that will likely be valuable to the AML research community. However, it has three critical issues: (1) no validation demonstrating data realism, (2) arbitrary modeling choices without proper justification, and (3) limited technical novelty. In its current state, it might be better suited for a domain-specific venue or applications track where tool contributions are more highly valued.

**Strengths:**

- The paper tackles a genuine barrier in AML research - the scarcity of publicly available transaction datasets due to privacy and regulatory constraints. This data scarcity significantly limits academic research and development of new detection methods.

- The paper provides substantial improvements over existing simulators.

- The framework offers two operation modes (knowledge-free and data-informed), making it accessible to researchers without proprietary data while allowing financial institutions to calibrate simulations to internal data.

- The open-source release with multiple pre-generated datasets, detailed documentation in appendices, supports reproducibility and community adoption.

**Weaknesses:**

W1: The paper's central claim is generating "realistic, configurable transaction data," but there is no empirical evidence demonstrating that synthetic data reflects real-world AML patterns:
- No comparison of statistical properties (degree distributions, transaction amounts, temporal patterns) between synthetic and real data;
- No fidelity metrics (e.g., Maximum Mean Discrepancy, Wasserstein distance, or feature distribution comparisons);
- No demonstration that models trained on synthetic data transfer to real data or vice versa.

Without validation, the claims of realism cannot be substantiated, and the utility of the synthetic data for developing practical AML systems remains unclear.

W2: The simulation involves numerous design decisions that appear arbitrary and lack empirical or theoretical grounding. For example:
- Logarithmic distribution for repeated laundering participation
- Spending probability based on sigmoid function
- Why use truncated Gaussian distributions for transaction amounts?
- Fan-in and fan-out patterns appear in both normal and alert categories (Figures 8-9). This makes ground truth labels fundamentally ambiguous

The paper reads like an engineering exercise combining various constraints rather than a principled data generation model. More justification from AML domain literature or empirical observations is needed.

W3: While the engineering effort is substantial, the core technical contributions are incremental. The agent-based simulation largely extends AMLSIM with more typologies and better accounting; the Bayesian optimization for hyperparameter tuning is standard practice; Feature engineering (Table 1) involves straightforward aggregation of transaction statistics. Currently, the contribution is primarily a software tool, which may be more suitable for domain-specific conferences, or applications tracks.

Minor issues:

- In fig. 7, colors are too similar across models, making visual comparison difficult. No error bars or confidence intervals are shown despite averaging over 10 seeds.

- The framework has numerous hyperparameters (degree distribution parameters, transaction amount distributions, typology frequencies, timing schemes). How sensitive are the experimental results to these choices?

- Real money laundering often involves tight timing constraints (rapid movement of funds to avoid detection) and amount conservation principles (money placed ≈ money integrated). The paper mentions timing schemes but does not clearly specify how these constraints can be enforced.

**Questions:**

See weaknesses.

---

### Note · Authors · 2025-11-12

I have read and agree with the venue's withdrawal policy on behalf of myself and my co-authors.